# On characterizing gender and locational composition of adult PLHIV in Nigeria: Implications for HIV programming

**Temitayo V. Lawal** [ORCID]*, **Oyewole K. Oyedele, Nifarta P. Andrew**

International Research Centre of Excellence, Institute of Human Virology Nigeria, Abuja, Nigeria

* latevi24@gmail.com

**Data Availability Statement:** Data used for this study is publicly available in the National report upload to the NAIIS website: https://www.naiis.ng/summary_sheet.

## Abstract

Human Immunodeficiency Virus (HIV) remains a global public health menace, and studies have suggested variations across demographic characteristics. This study attempted to characterize the gender and locational variations in the spread and control of HIV among adult Persons Living with HIV (PLHIV) to identify the clustering of PLHIV in Nigeria. We conducted an ecological analysis of data from the 2018 Nigeria HIV/AIDS Indicator and Impact Survey (NAIIS) report. To examine the statistical significance of differences between groups (by gender and location), chi-square and Fisher exact tests were conducted using the Real Statistics Resource Pack in Excel, and ArcGIS for visualization. Significance level was set at 5%. Overall, prevalence of HIV among adult PLHIV was 1.4%—the prevalence was slightly higher among women (1.8%) compared to men (1.0%). About 3/10 (29.3%) self-reported having received HIV test results. In the past 12 months before the survey, only 10.0% self-reported HIV testing among the population, with a significant difference (p<0.001) observed in the urban and rural area (12.4% in urban compared to 8.2% in rural). Another significant finding was the difference in the timing of HIV diagnosis between urban and rural areas (p<0.001). Adult PLHIV residing in rural areas had a higher rate of early detection (94.1%) compared to those in urban areas (70.3%). About 40.0% of HIV positive mothers reported awareness of their HIV status–approximately 58.0% of mothers living in the urban were aware of their HIV status compared to 29.8% in the rural area. The concentration of HIV was disproportionately higher in the rural populations compared to the urban. Findings from this study also show gender-disparities in HIV prevalence, diagnosis, and treatment. Interventions to bridge this gap should be promptly embraced in order to effectively combat the HIV epidemic in Nigeria and achieve an AIDS-free generation.

## Introduction

Globally, Human Immunodeficiency Virus (HIV) remains a public health menace which has infected 85.6 million [64.8 million–113.0 million] people since the start of the epidemic in 1981 [1]. Nearly two thirds of the world's new HIV infections originate in the African Region, with 25.7 million individuals living with HIV in sub-Saharan Africa; though it reduces by 33%

**Funding:** The authors received no specific funding for this work.

**Competing interests:** The authors have declared that no competing interests exist.

yet the most afflicted continent, resulting in 1.1 million persons infected in 2018 [1, 2]. Nigeria is the fourth most affected country with HIV prevalence of 1.4% in individuals 15–49 years—an approximate of 1.9 million people living with HIV (PLHIV) in 2019 [3, 4].

The HIV epidemic in Nigeria remains a mixed epidemic and is being driven by people who engage in low-risk sex (cohabiting or married sexual partners), whereas high-risk populations significantly raise the number of new infections [3, 5]. Behavioral characteristics and conditions associated with increase in the prevalence of HIV in Nigeria are seen amongst significant key populations such as female sex workers, (FSW), men who have sex with men (MSM) and people who inject drugs (PWID) [5, 6]. However, data has shown a decline in prevalence especially among FSW and PWID, but an increase in prevalence in MSM [5].

The present HIV testing and treatment target (95-95-95) aims to eradicate AIDS by 2030 by ensuring 95% of people living with HIV knowing their HIV status; 95% of people who know their status on treatment; and 95% of people on treatment with suppressed viral loads [7, 8]. Thus, the national HIV prevention program strategically focuses on lowering the incidence of new HIV infections in Nigeria with a three-pronged strategy to combining preventive services —biomedical, behavioral, and structural [9, 10]. This approach is in line with the Global HIV Prevention Coalition's (GPC) roadmap [11]. The aim is to guarantee that 90% of the population, including vulnerable and important populations, has access to HIV combination preventive programs by 2030 [8]. Another targeted HIV program in Nigeria is the 90% coverage of antiretroviral therapy (ART) among HIV positive women by 2025, which is essential for the detection and prevention of mother to child transmission of HIV (PMTCT) before pregnancy among reproductive age [12, 13].

However, risky behavioral practices and conditions increase a person's risk of contracting HIV. These include: sexually transmitted infections (STIs) from condom-free anal or vaginal sex; adverse consumption of alcohol and drugs during sexual behavior; sharing contaminated needles, syringes and other injecting equipment as well as drug solutions when injecting; receiving unsafe injections, blood transfusions, tissue transplants, unsterile cutting or piercing objects used for scarifications, tattoos; and other medical procedures and experiencing accidental needle stick injuries, including among health workers [14–16].

A prominent mode of transmission of HIV is from an HIV positive mother to her child [12]. Mother-to-child transmission (MTCT) is the primary means of HIV infection for most children under the age of 15 who are infected which could happen during nursing, labor and delivery, or pregnancy, and it accounts for a fairly high proportion of the infections among adolescents aged 10 to 19 years [17]. Additionally, shifting risk perceptions and social trends like the "Marlians," online dating, and transactional sexual behavior which could increase the rate of HIV transmission [18, 19].

In Nigeria, the HIV epidemic has an impact on all age groups and geographical locations. According to the report by National HIV and AIDS strategic framework in the country, female prevalence (15–49 years) is projected at 1.7% (1.6%–1.9%), substantially higher than male prevalence (15–49 years), which is estimated at 0.8% (0.7%–0.9%) [20]. The prevalence of this condition varied throughout States and Regions, with the South-South having the highest frequency (3.1%) and the North-West having the lowest (0.6%) [20].

Literatures opines on the prevalence, transmission, prevention, and risk factors of HIV in Nigeria [3, 6, 10, 12, 13]. However, little has been documented on the combined gender and locational composition of HIV indicators particularly based on the findings from a representative national survey. Hence, we characterized the regional and sex disparity in HIV prevalence, co-infections, testing and antiretroviral therapy (ART) retention among adult PLHIV to identify the clustering of PLHIV in Nigeria, determine the most affected gender and state and assess their associations. The goal is to inform on a programming guide to strengthen

interventional response targeting the most affected population to reduce the HIV burden and eradicate the transmission of AIDs towards achieving the 95-95-95 target.

## Methodology

### Data source, design and area

Publicly available (secondary) data from the Nigeria HIV/AIDS Indicator and Impact Survey 2018 Technical Report was used in this study. Specifically, the technical report of the survey was assessed, and data was manually extracted from it. The survey was a Population-based cross-sectional study on HIV Impact Assessment (PHIA) aimed at measuring national and regional HIV-related indicators, such as progress towards the UNAIDS 90-90-90 targets, to inform policy and funding decisions. This study adopted an ecological approach, which was partly inspired by the success of a similar attempt by previous researchers in investigating and supporting health outcomes at country-level [21, 22].

### Participants and sampling procedure

NAIIS survey included the de facto household population, consisting of individuals who slept in the household on the night prior to the interview. This group was representative of men and women of all ages who are residents of Nigeria. The survey was conducted across all 36 states and the Federal Capital Territory (FCT) of Nigeria (Fig 1) between 2018 and 2019, and it boasted a sample size of over 220,000 individuals, making it one of the largest HIV surveys conducted globally. For this study, we limited the participants to only those aged 15–64 years.

NAIIS sampled the population using a two-stage cluster sampling technique, selecting enumeration areas (EAs) followed by households. The sampling frame consisted of 662,855 EAs, a total of 28,900,478 households and 140,431,798 persons based on the 2006 Census, with an average of 44/212 households to persons ratio per EA. The EAs were mutually exclusive (non-overlapping). This ensured that all households and residents had an equal chance of being included in the survey.

All information on the data set and sampling methods can be found at https://www.naiis. ng/resources.

### Definition of variables

- HIV Prevalence was defined as the number of adults aged 15–64 years who tested positive to HIV using a serological rapid diagnostic testing algorithm based on Nigeria's National HIV Testing Guidelines, with laboratory confirmation of seropositive specimens using a supplemental assay.

- Self-reported HIV Testing: Number of participants who ever received an HIV test result and received an HIV test result.

- Timing of HIV Diagnosis: This was defined as the PLHIV who tested HIV positive in NAIIS but self-reported HIV negative. PLHIV who had no detectable antiretrovirals and who had a CD4 cell count <200 cells/μL were deemed to have "Very Late Diagnosis" and < 350 cells/μL were defined as having "Late Diagnosis".

- Retention of ART: PLHIV who self-reported still on ART after initiation ≥12 months prior to the survey.

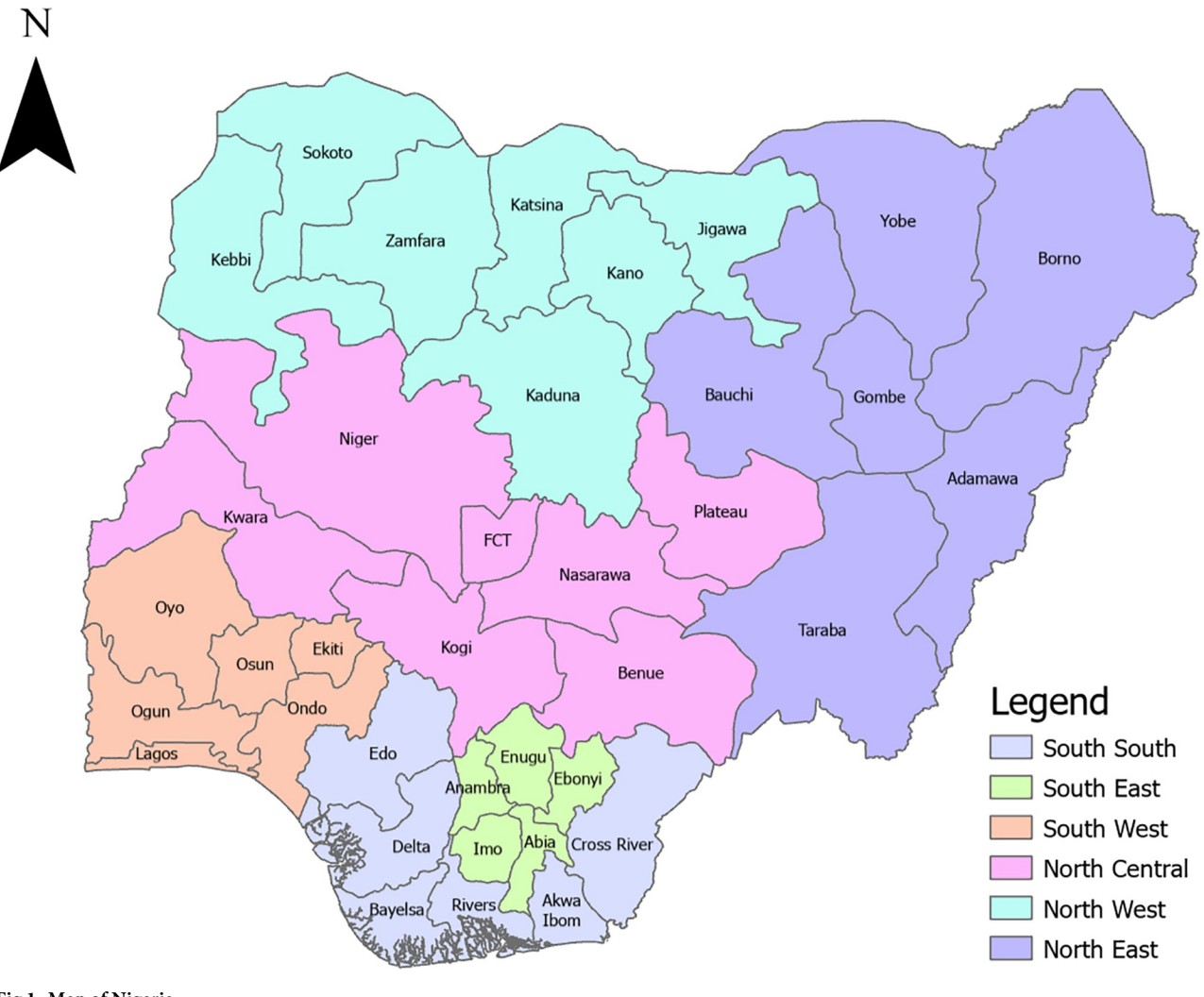

**Fig 1. Map of Nigeria.**

## Statistical procedures and mapping

All results presented in the technical report were based on weighted estimates. Analysis weights accounted for sample selection probabilities and adjusted for nonresponse and non-coverage. The nonresponse and noncoverage adjustment cells were constructed using the Chi-square Automatic Interaction Detector (CHAID) algorithm.

The prevalence rates and other study-level data were estimated from the NAIIS 2018 report. The report presented data in the form of percentages and total sample sizes, requiring the estimation of the sample size for each category. The extracted data was entered into Excel and cleaned before analysis. Further information on the data can be found at www.ciheb.org/PHIA and www.naiis.ng.

The data was dichotomized by place of residence (distinguishing between individuals residing in urban and rural areas)–as defined by NAIIS. In order to assess the statistical significance of the quantified "observed and predicted differences" between these groups, chi-square and Fisher exact tests were employed. Specifically, the Real Statistics Resource Pack was utilized as an analytical tool, which provides an array of statistical functions. All statistical analysis was

conducted in Excel and STATA version 18, and statistical significance was set at 5% (with starred p-values).

For mapping, the ArcGIS software was used to generate maps. The basemap shapefile was plotted onto the Nigeria administrative data extracted from the GADM database (https://gadm.org/download_country.html).

### Ethical considerations

Written and signed informed consent were obtained from each participant prior to commencement of the study. For PLHIV < 18 years, consent was sought from the household head. The study posed minimal risk to the participants, and all information was collected anonymously and held confidentially. NAIIS was approved by the Institutional Review Board of the University of Maryland, Baltimore and the U.S. Centers for Disease Control (CDC) in Atlanta, GA, USA (protocol #7103) and the Nigerian National Health Research Ethics Committee. However, since this study involved secondary analysis of publicly available data from the NAIIS Technical Report, it did not require ethical approval. The data was accessed without further permission, as it was already publicly available.

## Results

### Demographic characteristics of participants

More than half (57.0%) of the surveyed households resided in rural areas. Observably, some states had higher rural population than others indicating that the rural-urban divide varies from state to state. Among the de facto household population aged 15–64 years used in this study, 45.6% were men and 54.4% were women. Also, 42.0% were residing in the rural areas, while 58.0% were resident in the urban areas (Table 1).

### HIV prevalence and testing

The prevalence of HIV among the population was 1.4%. The prevalence was slightly higher among women (1.8%) compared to men (1.0%). Across residence, the study found a statistically significant difference between HIV prevalence rate among the women and among the entire population (p-value < 0.001 for both categories). Across gender, HIV prevalence rate

**Table 1. Demographic characteristics of participants.**

| Variable | Sample<br>n (col %) |
|---|---|
| **Gender** | |
| Male | 94444 (45.6) |
| Female | 112557 (54.4) |
| **Place of residence** | |
| Urban | 86940 (42.0) |
| Rural | 120061 (58.0) |
| **Age group** | |
| 15–24 years | 63549 (30.7) |
| 25–34 years | 54648 (26.4) |
| 35–44 years | 41400 (20.0) |
| 45–54 years | 27945 (13.5) |
| 55–64 years | 19459 (9.4) |
| **Overall** | 207001 (100.0) |

was significantly higher among females (1.8%) than males (1.0%) (p-value < 0.001) (Tables 2 and 3).

Fig 2 shows the percentages of households with at least one HIV-positive member, with variations observed across states and place of residence. Notably, Akwa-Ibom and Benue had the highest percentage of households with at least one HIV-positive member (9.4%), followed by Taraba (7.1%), and Rivers (7.0%). On the other hand, Katsina (0.6%), Zamfara (0.7%), and Yobe, Sokoto and Jigawa (0.8%) had the lowest percentage of households with at least one HIV-positive member. One notable thing to note is that Akwa-Ibom and Benue had the highest rural percentage of households with at least one HIV-positive member (9.8% and 9.4%), while Katsina and Jigawa had the lowest (0.5% and 0.6% respectively).

Fig 3 presents the prevalence rate of HIV among participants. Prevalence of HIV ranged between 0.3% and 4.8% across states. The prevalence rate was high in Taraba, Benue, Anambra, Rivers, and Akwa-Ibom states, and lowest in the Norther region (including Kebbi, Sokoto, Zamfara, Katsina, Niger, Kano, Jigawa, Yobe, and Bauchi).

Previous HIV screening was low among participants; 29.3% self-reported ever having received HIV test results. Disaggregation by place of residence showed that self-reported HIV testing was significantly higher in the urban area (36.8%) compared to the rural area (23.8%) (p-value < 0.001). In the past 12 months before the survey, only 10.0% self-reported HIV testing among the population, with a significant difference observed in the urban and rural area (12.4% in the urban area compared to 8.2% in the rural area). Similarly, the study showed a significant difference in self-reported HIV testing among men and women in rural and urban areas (32.8; 23.0 and 41.0; 24.7 respectively), and (11.2%; 8.0 and 13.6%; 8.5% respectively) 12 months before the survey. Similarly, disaggregation by gender revealed a statistically significant difference between self-reported HIV testing (27.7% among male and 32.6% among female) and self-reported HIV testing within 12 months before the survey (9.5% among male, and 10.9% among female) (Tables 2 and 3).

## HIV diagnosis, treatment, clinical perspective and co-infections of PLHIV

Tables 2 and 3 further shows the diagnostic, treatment, and clinical characteristics of PLHIV. Findings from this table showed that there was statistically significant difference between awareness HIV status among PLHIV resident in the urban (32.6%) and rural (26.0%) areas. Among PLHIV who were aware of their HIV positivity status, 89.8% were on antiretroviral therapy–this was not significantly different by residence (89.4% in the urban, and 90.0% in the rural areas). This was also similar to the findings on the awareness of HIV status among women– 35.6% were aware of their HIV positivity status in the urban area while 25.4% reported awareness in the rural area. Among all participants, viral load suppression was significantly different across urban and rural divide. More participants in the urban area (46.7%) had viral load suppression compared to the rural area (40.3%). This statistically significant difference was also observed among female PLHIV (51.1% had viral load suppression in the urban area compared to 41.2% in the rural area).

There was a significant difference in the timing of HIV diagnosis between urban and rural areas, with a higher proportion (16.0%) of participants experiencing a late diagnosis. Among people living with HIV (PLHIV), those residing in rural areas had a higher rate of early detection (94.1%) compared to those in urban areas (70.3%). The prevalence of Hepatitis C, co-infected with HIV, was also significantly different between rural (1.8%) and urban (0.4%) PLHIV. These findings were consistent across both genders and across rural and urban areas (p-value < 0.001).

**Table 2. HIV prevalence and clinical diagnosis among adult PLHIV by residence.**

| Variable | Sample n (col %) | Place of residence | | $\chi^2$ (p-value) |
|---|---|---|---|---|
| | | Urban n (col %) | Rural n (col %) | |
| **HIV PREVALENCE** | | | | |
| **HIV Prevalence among men** | | | | |
| Has HIV | 748 (1.0) | 290 (0.9) | 458 (1.0) | 1.935 |
| Does not have HIV | 77222 (99.0) | 31882 (99.1) | 45340 (99.0) | (0.164) |
| **HIV Prevalence among women** | | | | |
| Has HIV | 1697 (1.8) | 650 (1.6) | 1047 (1.9) | 12.005 |
| Does not have HIV | 94049 (98.2) | 39968 (98.4) | 54081 (98.1) | ($<0.001$) * |
| **HIV Prevalence in the population** | | | | |
| Has HIV | 2460 (1.4) | 946 (1.3) | 1514 (1.5) | 12.175 |
| Does not have HIV | 171256 (98.6) | 71844 (98.7) | 99412 (98.5) | ($<0.001$) * |
| **SELF REPORTED HIV TESTING** | | | | |
| **Self-reported HIV Testing** | | | | |
| Ever received HIV test result | 53054 (29.3) | 28169 (36.8) | 24885 (23.8) | 3605.785 |
| Never received HIV test result | 128053 (70.7) | 48378 (63.2) | 79675 (76.2) | ($<0.001$) * |
| **Self-reported HIV Testing 12 months before survey** | | | | |
| Ever received HIV test result | 17493 (10.0) | 9120 (12.4) | 8373 (8.2) | 841.016 |
| Never received HIV test result | 158158 (90.0) | 64426 (87.6) | 93732 (91.8) | ($<0.001$) * |
| **Self-reported HIV Testing among men** | | | | |
| Ever received HIV test result | 22175 (27.1) | 11168 (32.8) | 11007 (23.0) | 967.845 |
| Never received HIV test result | 59732 (72.9) | 22881 (67.2) | 36851 (77.0) | ($<0.001$) * |
| **Self-reported HIV Testing 12 months before survey among men** | | | | |
| Ever received HIV test result | 7457 (9.3) | 3703 (11.2) | 3754 (8.0) | 235.022 |
| Never received HIV test result | 72539 (90.7) | 29362 (88.8) | 43177 (92.0) | ($<0.001$) * |
| **Self-reported HIV Testing among women** | | | | |
| Ever received HIV test result | 31429 (31.7) | 17424 (41.0) | 14005 (24.7) | 2981.905 |
| Never received HIV test result | 67771 (68.3) | 25074 (59.0) | 42697 (75.3) | ($<0.001$) * |
| **Self-reported HIV Testing 12 months before survey among women** | | | | |
| Ever received HIV test result | 10195 (10.7) | 5505 (13.6) | 4690 (8.5) | 637.447 |
| Never received HIV test result | 85460 (89.3) | 34976 (86.4) | 50484 (91.5) | ($<0.001$) * |
| **HIV DIAGNOSIS AND TREATMENT AMONG HIV POSITIVE CLIENTS** | | | | |
| **Awareness of HIV Status** | | | | |
| Aware of HIV positivity | 761 (28.6) | 341 (32.6) | 420 (26.0) | 13.446 |
| Unaware of HIV positivity | 1899 (71.4) | 705 (67.4) | 1194 (74.0) | ($<0.001$) * |
| **HIV Treatment Status** | | | | |
| On ART | 683 (89.8) | 305 (89.4) | 378 (90.0) | 0.064 |

*(Continued)*

**Table 2.** (Continued)

| Variable | Sample n (col %) | Place of residence | | $\chi^2$ (p-value) |
| --- | --- | --- | --- | --- |
| | | Urban n (col %) | Rural n (col %) | |
| Not on ART | 78 (10.2) | 36 (10.6) | 42 (10.0) | (0.801) |
| **Awareness of HIV Status among men** | | | | |
| Aware of HIV positivity | 225 (27.2) | 85 (27.4) | 140 (27.0) | 0.015 |
| Unaware of HIV positivity | 603 (72.8) | 225 (72.6) | 378 (73.0) | (0.902) |
| **HIV Treatment Status among men** | | | | |
| On ART | 211 (93.8) | 81 (95.3) | 130 (92.9) | 0.538 |
| Not on ART | 14 (6.2) | 4 (4.7) | 10 (7.1) | (0.463) |
| **Awareness of HIV Status among women** | | | | |
| Aware of HIV positivity | 540 (29.5) | 262 (35.6) | 278 (25.4) | 22.180 |
| Unaware of HIV positivity | 1292 (70.5) | 474 (64.4) | 818 (74.6) | (< 0.001) * |
| **HIV Treatment Status among women** | | | | |
| On ART | 474 (87.8) | 229 (87.4) | 245 (88.1) | 0.066 |
| Not on ART | 66 (12.2) | 33 (12.6) | 33 (11.9) | (0.797) |
| **Viral Load** | | | | |
| < 1000 copies/mL | 1172 (42.8) | 503 (46.7) | 669 (40.3) | 10.882 |
| ≥ 1000 copies/mL | 1567 (57.2) | 575 (53.3) | 992 (59.7) | (< 0.001) * |
| **Viral Load among men** | | | | |
| < 1000 copies/mL | 328 (38.8) | 124 (38.9) | 204 (38.8) | 0.001 |
| ≥ 1000 copies/mL | 517 (61.2) | 195 (61.1) | 322 (61.2) | (0.980) |
| **Viral Load among women** | | | | |
| < 1000 copies/mL | 856 (45.2) | 388 (51.1) | 468 (41.2) | 17.948 |
| ≥ 1000 copies/mL | 1038 (54.8) | 371 (48.9) | 667 (58.8) | (< 0.001) * |
| **CLINICAL PERSPECTIVES ON PLHIV** | | | | |
| **CD4 count** | | | | |
| < 500 cells/μL | 1330 (49.2) | 515 (48.3) | 815 (49.8) | 0.621 |
| ≥ 500 cells/μL | 1373 (50.8) | 552 (51.7) | 821 (50.2) | (0.431) |
| **CD4 count among men** | | | | |
| < 500 cells/μL | 490 (58.7) | 187 (59.6) | 303 (58.2) | 0.158 |
| ≥ 500 cells/μL | 345 (41.3) | 127 (40.4) | 218 (41.8) | (0.691) |
| **CD4 count among women** | | | | |
| < 500 cells/μL | 820 (43.9) | 317 (42.1) | 503 (45.1) | 1.658 |
| ≥ 500 cells/μL | 1048 (56.1) | 436 (57.9) | 612 (54.9) | (0.198) |
| **Timing of HIV Diagnosis** | | | | |
| Early | 399 (84.0) | 142 (70.3) | 257 (94.1) | 117.984 |
| Late | 34 (7.2) | 86 (42.6) | - | (<0.001) * |
| Very late | 42 (8.8) | 26 (12.9 | 16 (5.9) | |
| **Timing of HIV Diagnosis among men** | | | | |
| Early | 100 (65.8) | 40 (66.7) | 60 (65.2) | 5.228 |
| Late | 40 (26.3) | 12 (20.0) | 28 (30.4) | (0.073) |
| Very late | 12 (7.9) | 8 (13.3) | 4 (4.3) | |
| **Timing of HIV Diagnosis among women** | | | | |
| Early | 237 (73.4) | 103 (72.5) | 134 (74.0) | 4.161 |
| Late | 54 (16.7) | 20 (14.1) | 34 (18.8) | (0.125) |
| Very late | 32 (9.9) | 19 (13.4) | 13 (7.2) | |
| **Retention of ART** | | | | |

*(Continued)*

**Table 2.** (Continued)

| Variable | Sample n (col %) | Place of residence | | $\chi^2$ (p-value) |
|---|---|---|---|---|
| | | Urban n (col %) | Rural n (col %) | |
| Yes | 569 (94.2) | 270 (94.1) | 299 (94.3) | 0.017 |
| No | 35 (5.8) | 17 (5.9) | 18 (5.7) | (0.898) |
| **Retention of ART among men** | | | | |
| Yes | 174 (96.1) | 73 (98.6) | 101 (94.4) | 2.132 |
| No | 7 (3.9) | 1 (1.4) | 6 (5.6) | (0.144) |
| **Retention of ART among women** | | | | |
| Yes | 395 (93.4) | 197 (92.5) | 198 (94.3) | 0.553 |
| No | 28 (6.6) | 16 (7.5) | 12 (5.7) | (0.457) |
| **PMTCT TRANSMISSION** | | | | |
| **Knowledge of HIV Status among HIV positive mothers who gave birth within 12months** | | | | |
| Known | 3382 (40.4) | 1842 (57.7) | 1540 (29.8) | 637.657 |
| Unknown | 4980 (59.6) | 1351 (42.3) | 3629 (70.2) | (< 0.001) * |
| **HIV Treatment** | | | | |
| Received antiretrovirals | 108 (84.4) | 50 (79.4) | 58 (89.2) | 2.362 |
| Did not receive antiretrovirals | 20 (15.6) | 13 (20.6) | 7 (10.8) | (0.124) |
| **HIV CO-INFECTIONS** | | | | |
| **Hepatitis B** | | | | |
| Present | 847 (8.1) | 338 (7.6) | 509 (8.5) | 2.842 |
| Absent | 9591 (91.9) | 4114 (92.4) | 5477 (91.5) | (0.092) |
| **Hepatitis B among men** | | | | |
| Present | 457 (10.4) | 178 (9.8) | 279 (10.8) | 1.102 |
| Absent | 3937 (89.6) | 1634 (90.2) | 2303 (89.2) | (0.294) |
| **Hepatitis B among women** | | | | |
| Present | 349 (5.8) | 145 (5.5) | 204 (6.0) | 0.685 |
| Absent | 5695 (94.2) | 2495 (94.5) | 3200 (94.0) | (0.408) |
| **Hepatitis C** | | | | |
| Present | 126 (1.2) | 18 (0.4) | 108 (1.8) | 41.970 |
| Absent | 10313 (98.8) | 4435 (99.6) | 5878 (98.2) | (< 0.001) * |
| **Hepatitis C among men** | | | | |
| Present | 59 (1.3) | 13 (0.7) | 46 (1.8) | 9.113 |
| Absent | 4336 (98.7) | 1800 (99.3) | 2536 (98.2) | (0.003) * |
| **Hepatitis C among women** | | | | |
| Present | 64 (1.1) | 3 (0.1) | 61 (1.8) | 39.977 |
| Absent | 5980 (98.9) | 2637 (99.9) | 3343 (98.2) | (< 0.001) * |

*—significant at 5%

Table 3 shows diagnostic, treatment and clinical characteristics of PLHIV, disaggregated by gender. The table showed a statistically significant difference between the percentage of PLHIV who are on HIV treatment (93.8% among male and 87.6% among female). A statistically significant difference between the CD4 count values between male (58.7%) and female (43.8%). Having an co-infection (Hepatitis B) was also significant among male (10.3%) and female (5.8%).

**Table 3. HIV Prevalence and clinical diagnosis among adult PLHIV by gender.**

| Variable | Sample n (col %) | Gender | | $\chi^2$ (p-value) |
|---|---|---|---|---|
| | | Male n (col %) | Female n (col %) | |
| **HIV PREVALENCE** | | | | |
| **HIV Prevalence in the population** | | | | |
| Has HIV | 2503 (1.4) | 780 (1.0) | 1723 (1.8) | 193.172 |
| Does not have HIV | 171213 (98.6) | 77190 (99.0) | 94023 (98.2) | (< 0.001) * |
| **SELF REPORTED HIV TESTING** | | | | |
| **Self-reported HIV Testing** | | | | |
| Ever received HIV test result | 55027 (30.4) | 22688 (27.7) | 32339 (32.6) | 509.277 |
| Never received HIV test result | 126080 (69.6) | 59219 (72.3) | 66861 (67.4) | (< 0.001) * |
| **Self-reported HIV Testing 12 months before survey** | | | | |
| Ever received HIV test result | 18026 (10.3) | 7600 (9.5) | 10426 (10.9) | 92.599 |
| Never received HIV test result | 158158 (90.0) | 72396 (90.5) | 85229 (89.1) | (< 0.001) * |
| **HIV DIAGNOSIS AND TREATMENT AMONG HIV POSITIVE CLIENTS** | | | | |
| **Awareness of HIV Status** | | | | |
| Aware of HIV positivity | 772 (29.0) | 224 (27.1) | 548 (29.9) | 2.264 |
| Unaware of HIV positivity | 1888 (71.0) | 604 (72.9) | 1284 (70.1) | (0.132) |
| **HIV Treatment Status** | | | | |
| On ART | 690 (26.0) | 210 (93.8) | 480 (87.6) | 6.438 |
| Not on ART | 82 (64.0) | 14 (6.3) | 68 (11.5) | (0.011) * |
| **Viral Load** | | | | |
| < 1000 copies/mL | 1190 (43.4) | 328 (38.8) | 862 (45.5) | 10.661 |
| ≥ 1000 copies/mL | 1549 (56.6) | 517 (61.2) | 1032 (54.5) | (0.001) * |
| **CLINICAL PERSPECTIVES ON PLHIV** | | | | |
| **CD4 count** | | | | |
| < 500 cells/μL | 1308 (48.4) | 490 (58.7) | 818 (43.8) | 51.246 |
| ≥ 500 cells/μL | 1395 (51.6) | 345 (41.3) | 1050 (56.2) | (< 0.001) * |
| **Timing of HIV Diagnosis** | | | | |
| Early | 337 (70.9) | 101 (66.4) | 236 (73.1) | 4.980 |
| Late | 94 (19.8) | 39 (25.7) | 55 (17.0) | (0.083) |
| Very late | 44 (9.3) | 12 (7.9) | 32 (9.9) | |
| **Retention of ART** | | | | |
| Yes | 569 (94.2) | 174 (96.1) | 395 (93.4) | 1.759 |
| No | 35 (5.8) | 7 (3.9) | 28 (6.6) | (0.185) |
| **HIV CO-INFECTIONS** | | | | |
| **Hepatitis B** | | | | |
| Present | 804 (7.7) | 453 (10.3) | 351 (5.8) | 72.539 |
| Absent | 9634 (92.3) | 3941 (89.7) | 5693 (94.2) | (< 0.001) * |
| **Hepatitis C** | | | | |
| Present | 117 (1.1) | 57 (1.3) | 60 (1.0) | 2.125 |
| Absent | 10322 (98.9) | 4338 (98.7) | 5984 (99.0) | (0.145) |

*—significant at 5%

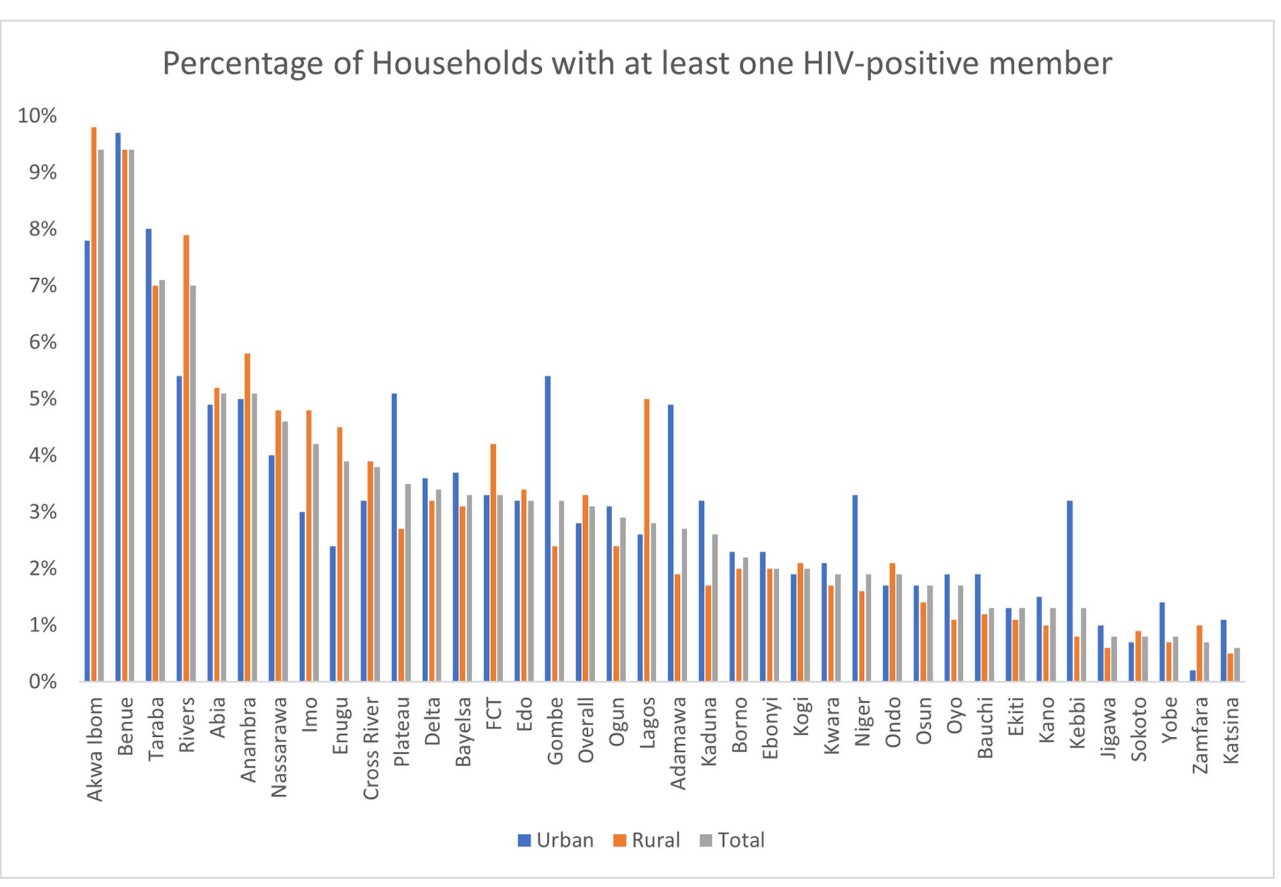

**Fig 2. Percentage of households with at least one HIV-positive member.**

## Prevention of mother to child transmission among HIV-positive mothers

About 40.0% of HIV positive mothers, who had given birth 12 months before the survey, were aware of their HIV status. The table shows a statistically significant difference across rural and urban areas. Approximately 58.0% of mothers who delivered within a year that were living in the urban were aware of their HIV status compared to 29.8% in the rural area. Also, 84.4% of all HIV positive mothers who delivered within a year were receiving antiretrovirals–this finding was almost similar across urban (79.4%) and rural (89.2%) areas.

## Discussion

The NAIIS survey is a comprehensive survey that provides useful insights into the prevalence and characteristics of HIV/AIDS in Nigeria. The survey is one of the largest HIV/AIDs datasets in the world, recruiting 83,909 heads of households across all states including the Federal Capital Territory [23, 24]. This large sample size would provide a more comprehensive understanding of the epidemiology of HIV/AIDS in Nigeria, which is critical for developing effective prevention and treatment strategies. More than half of the surveyed households were resident in rural areas—this is not surprising, as Nigeria is still largely an agrarian society, and a significant proportion of the population lives in rural areas [25]. Due to this imbalance, the findings from this study were weighted based on geographical composition to ensure robustness.

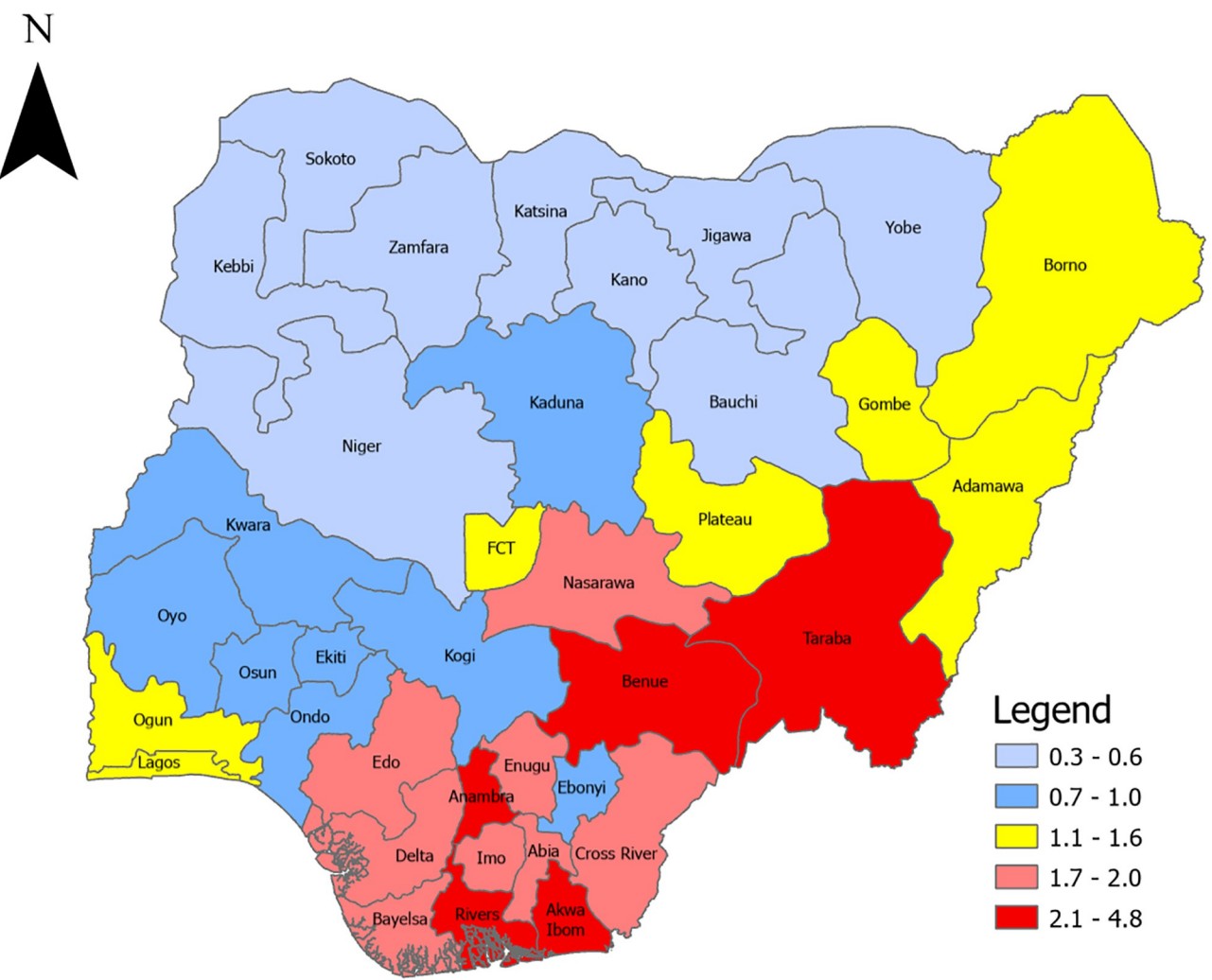

**Fig 3. HIV Prevalence among participants aged 15-64years by state.** Source: NAIIS 2018.

The outcome of the study showed that the prevalence of PLHIV was high in Taraba, Benue, Anambra, Rivers and Akwa Ibom states, and generally low in the Northern region. This finding on high prevalence is consistent with a literature that analyzed national HIV testing services data from October 2020 to September 2021 to derive state-level seropositivity rates which revealed a high rate in Benue (5.7%, 95% CI: 5.0–6.3), Rivers (5.2%, 95% CI: 4.6–5.8%), Akwa Ibom (3.5%, 95% CI: 2.9–4.1%), Edo (3.4%, 95% CI: 2.9–4.0%) and Taraba (3.0%, 95% CI: 2.6–3.7%) [26]. In order to properly explain these state-level differences in PLHIV prevalence, it may be necessary to conduct further sub-national disaggregated research to understand drivers of HIV prevalence in these states. It is also noteworthy that there are locational variations in the percentage of households with at least one HIV positive household member–while some states had higher rural population of households with HIV, others had higher urban population of households with HIV. It is strikingly obvious that Plateau, Gombe, Adamawa, Niger, Kebbi, and Taraba had high urban HIV household population while Lagos, FCT, Enugu, Zamfara, Rivers, Imo, had higher rural HIV household population. This could suggest disparities in intra-state HIV transmission dynamics.

The prevalence rate of HIV in Nigeria, as reported by the NAIIS survey, is a significant public health concern. Our findings on the overall adult prevalence of HIV was lower than the other estimates which reported between 1.5–2.8% [26–28]. However, our estimate of 1.4% is in agreement with the UNAIDS reported prevalence of HIV in Nigeria among adults in 2019 [29]. This prevalence indicates the effectiveness of massive investments in HIV prevention and treatment interventions in the country. Despite the improvement, HIV prevalence is still high compared to other countries as Nigeria still has the third highest HIV burden in the world due to her large population [29, 30]. Therefore, continued efforts are needed to reduce the burden of HIV in the country to end the HIV epidemic.

We found that the prevalence of HIV was higher among women than men, and this is consistent with previous research which has shown that women are disproportionately affected by HIV in Nigeria [28, 31]. This gender disparity could be due to a range of factors, including biological differences and gender-based inequalities that affect women's access to HIV prevention and treatment services [32, 33]. Although men are often more likely to engage in high-risk behaviors such as unprotected sex, injection drug use, and having multiple sexual partners which puts them at risk of contracting HIV, women are at a higher risk of contracting HIV from men due to a higher risk of exposure to infected semen of men with multiple sexual partners during heterosexual intercourse [34–36]. In addition, the lining of the vagina may also be more susceptible to tears and abrasions during intercourse, providing a potential entry point for the virus. If the high prevalence of HIV among women persists, there is a possibility of increased prevalence of HIV in future generations through mother to child transmission.

Furthermore, our study found a statistically significant difference in HIV prevalence rates across residence, with a higher prevalence rate among women living in rural areas. This finding could be due to a range of factors, including limited access to HIV prevention and treatment services in rural areas, lack of education on HIV prevention, and cultural factors that stigmatize HIV-positive individuals in rural areas. Rural areas often have limited healthcare resources, including HIV testing and treatment services which can lead to fewer people getting tested and receiving the care they need to manage their HIV status [37–39]. Rural areas also tend to have higher rates of poverty, which can lead to a range of health disparities, including higher rates of HIV infection as a result of limited access to education, healthcare, and other resources that can help prevent HIV transmission [37].

The findings of the study suggest that there are significant differences in the timing of HIV diagnosis and prevalence of co-infections between rural and urban areas. The proportion of participants experiencing an early detection of HIV was higher in the rural areas compared to those in urban areas. This is surprising as the rural areas is characterized by limited access to healthcare and lower levels of education and awareness about HIV prevention and transmission [37, 39, 40]. The higher prevalence of Hepatitis C among PLHIV in rural areas may also be related to limited access to healthcare (which makes testing and treatment more difficult), low education–which may be responsible for lower awareness of HIV status and delayed diagnosis, and higher rates of injection drug use in some rural communities [41, 42]. Plausibly, healthcare providers may tend to spend more time to spend with patients during appointments, which can facilitate conversations about HIV testing and prevention. Previous research has shown that primary care is an ideal setting to conduct HIV testing for those who have access to regular health care and healthcare providers in urban settings may have a higher patient load and less time for individualized care, which can lead to missed opportunities for HIV testing and prevention [43].

The differences in diagnostic, treatment, and clinical characteristics of PLHIV by gender were also detected. Higher percentage of male PLHIV on HIV treatment may be related to differences in healthcare-seeking behaviors or social stigma surrounding HIV among women

[44]. Higher percentage of female PLHIV with Immunosuppression (CD4 count value < 500) suggests that there may be gender-specific barriers to HIV diagnosis and treatment that need to be addressed. Although there is no clear evidence to suggest that immunosuppression is consistently higher among females, however, there are certain circumstances in which females may experience more immunosuppression than males [45]. One of these may be during pregnancy when a woman's immune system is naturally suppressed to prevent rejection of the fetus, and the presence of certain autoimmune diseases, such as lupus and multiple sclerosis, which are more common in women than men [45, 46].

Findings from this study revealed that majority of HIV positive mothers were unaware of their status, which can have serious implications for both their own health and the health of their children. When a mother is not aware of her HIV-positive status, she is less likely to receive appropriate medical care and antiretroviral therapy (ART) during pregnancy, delivery, and breastfeeding–this increases the risk of mother-to-child transmission of HIV [47–49]. A significant percentage of mothers who tested positive for HIV and gave birth within a year were undergoing antiretroviral treatment. While this is a positive development, further efforts are necessary to put an end to the HIV epidemic and accomplish the ultimate objective of creating an AIDS-free generation.

## Conclusion and recommendation

Understanding the gender and locational dynamics of the epidemic is crucial in developing targeted interventions that can effectively address the needs of PLHIV. This is because drivers of the HIV epidemic in urban and rural areas may differ, and interventions that are effective in one setting may not be effective in another. While the survey's finding of a lower overall prevalence rate than previous estimates is promising, the high prevalence rate among women and those living in rural areas remains a significant public health concern. Findings from this study highlights the need for continued efforts to address the HIV epidemic in Nigeria, with a focus on targeted interventions that address the unique needs and challenges faced by different populations, particularly women and those living in rural areas. By taking a gender-specific and location-specific approach, stakeholders and policymakers can develop more effective interventions that have the potential to reduce the burden of HIV in Nigeria.

## Study strength and limitation

NAIIS is a nationally representative data which incorporated rigorous design method. We leverage the 2018 NAIIS data presented in the comprehensive report of the survey. Therefore, the results of our analysis are as good as the original study (NAIIS) from which the data was obtained. However, restricted access to variable-level data, and other co-infections like Tuberculosis limited variable inclusion and further analysis.

## Acknowledgments

The study investigators express their gratitude to the Federal Ministry of Health, collaborating institutions, and donors who supported the NAIIS 2018 survey, for providing the great quantity of data summarized in the report we have adopted for this analysis.

## Author Contributions

**Conceptualization:** Temitayo V. Lawal, Oyewole K. Oyedele.

**Data curation:** Temitayo V. Lawal, Oyewole K. Oyedele.

**Formal analysis:** Temitayo V. Lawal, Oyewole K. Oyedele, Nifarta P. Andrew.

**Investigation:** Temitayo V. Lawal, Oyewole K. Oyedele.

**Methodology:** Temitayo V. Lawal, Oyewole K. Oyedele, Nifarta P. Andrew.

**Validation:** Temitayo V. Lawal, Oyewole K. Oyedele, Nifarta P. Andrew.

**Writing – original draft:** Temitayo V. Lawal, Oyewole K. Oyedele, Nifarta P. Andrew.

**Writing – review & editing:** Temitayo V. Lawal, Oyewole K. Oyedele, Nifarta P. Andrew.

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
