## [Decision Letter · Decision Letter 0]

21 Dec 2023

PGPH-D-23-02288

On Characterizing Gender and Locational Composition of Adult PLHIV in Nigeria: implications for HIV programming

Dear Lawal,

Thank you for submitting your manuscript to PLOS Global Public Health. After careful consideration, we feel that it has merit but does not fully meet PLOS Global Public Health’s publication criteria as it currently stands. Therefore, we invite you to submit a revised version of the manuscript that addresses the points raised during the review process.

We look forward to receiving your revised manuscript.

Kind regards,

Collins Otieno Asweto, PhD

Academic Editor

Journal Requirements:

Additional Editor Comments (if provided):

Reviewers' comments:

Reviewer's Responses to Questions

**Comments to the Author**

1. Does this manuscript meet PLOS Global Public Health’s publication criteria? Is the manuscript technically sound, and do the data support the conclusions? The manuscript must describe methodologically and ethically rigorous research with conclusions that are appropriately drawn based on the data presented.

Reviewer #1: Yes

Reviewer #2: Yes

2. Has the statistical analysis been performed appropriately and rigorously?

Reviewer #1: Yes

Reviewer #2: Yes

3. Have the authors made all data underlying the findings in their manuscript fully available (please refer to the Data Availability Statement at the start of the manuscript PDF file)?

Reviewer #1: Yes

Reviewer #2: Yes

4. Is the manuscript presented in an intelligible fashion and written in standard English?

Reviewer #1: Yes

Reviewer #2: Yes

5. Review Comments to the Author

Reviewer #1: Overall, a good study and analysis of a major nation-wide survey.

The findings on urban/rural and among women have important implications for organizing the national HIV/AIDS response and as such merits publication with no changes

Reviewer #2: I would like to congratulate all authors for undertaking this important project.

The topic is quite relevant in the contemporary contexts: We see gaps in HIV prevalence, treatment and care in the contexts of gender and residence.

Authors are required to address the following issues in order to improve the quality of their papers

General: The must be concise, improve grammars and ensure coherence, hence smooth flow from one sentence and one paragraph to another.

Background: More contexts on gender and spatial relationship must be provided for readers to establish clearly the need for this important study.

Methods: Information provided on type of analysis that were taken are not enough.

Results: First paragraph and table - The proportion of rural residents/participants on the texts and tables are not tallying with each other. Pg15/16 - The word North region in the sentence implies that, all other states were compared against those in that region. It is however not clear, from the text alone, where these states come from. Better to put a region for them or omit the word Northern region.

Discussion: A statement that a large proportion of women have immunosuppression (low CD4 count) is wrong as the findings suggest otherwise and particularly when they are compared with men. Discussion involves explaining study's findings and in most cases, the explanation must be in the contexts of existing evidence. Comparing with other studies was largely missing.

References: The list is exhaustive. Authors are urged to make few amendments to make it even better. Reference number 11 is wrongly cited.

6. PLOS authors have the option to publish the peer review history of their article (what does this mean?). If published, this will include your full peer review and any attached files.

**Do you want your identity to be public for this peer review?** For information about this choice, including consent withdrawal, please see our Privacy Policy.

Reviewer #1: **Yes: **Lakshmi N Balaji

Reviewer #2: No

---

## [Decision Letter · Decision Letter 1]

14 Feb 2024

PGPH-D-23-02288R1

On Characterizing Gender and Locational Composition of Adult PLHIV in Nigeria: implications for HIV programming

Dear Lawal,

Thank you for submitting your manuscript to PLOS Global Public Health. After careful consideration, we feel that it has merit but does not fully meet PLOS Global Public Health’s publication criteria as it currently stands. Therefore, we invite you to submit a revised version of the manuscript that addresses the points raised during the review process.

We look forward to receiving your revised manuscript.

Kind regards,

Collins Otieno Asweto, PhD

Academic Editor

Journal Requirements:

2. Please provide separate figure files in .tif or .eps format only and remove any figures embedded in your manuscript file. Please also ensure all files are under our size limit of 10MB.

3. Some material included in your submission may be copyrighted. According to PLOS’s copyright policy, authors who use figures or other material (e.g., graphics, clipart, maps) from another author or copyright holder must demonstrate or obtain permission to publish this material under the Creative Commons Attribution 4.0 International (CC BY 4.0) License used by PLOS journals. Please closely review the details of PLOS’s copyright requirements here: PLOS Licenses and Copyright. If you need to request permissions from a copyright holder, you may use PLOS's Copyright Content Permission form.

Potential Copyright Issues:

Figs 1&3: please (a) provide a direct link to the base layer of the map (i.e., the country or region border shape) and ensure this is also included in the figure legend; and (b) provide a link to the terms of use / license information for the base layer image or shapefile. We cannot publish proprietary or copyrighted maps (e.g. Google Maps, Mapquest) and the terms of use for your map base layer must be compatible with our CC-BY 4.0 license. 

"

Additional Editor Comments (if provided):

Reviewers' comments:

Reviewer's Responses to Questions

**Comments to the Author**

1. If the authors have adequately addressed your comments raised in a previous round of review and you feel that this manuscript is now acceptable for publication, you may indicate that here to bypass the “Comments to the Author” section, enter your conflict of interest statement in the “Confidential to Editor” section, and submit your "Accept" recommendation.

Reviewer #3: (No Response)

Reviewer #4: (No Response)

2. Does this manuscript meet PLOS Global Public Health’s publication criteria? Is the manuscript technically sound, and do the data support the conclusions? The manuscript must describe methodologically and ethically rigorous research with conclusions that are appropriately drawn based on the data presented.

Reviewer #3: Yes

Reviewer #4: Partly

3. Has the statistical analysis been performed appropriately and rigorously?

Reviewer #3: Yes

Reviewer #4: Yes

4. Have the authors made all data underlying the findings in their manuscript fully available (please refer to the Data Availability Statement at the start of the manuscript PDF file)?

Reviewer #3: Yes

Reviewer #4: Yes

5. Is the manuscript presented in an intelligible fashion and written in standard English?

Reviewer #3: Yes

Reviewer #4: Yes

6. Review Comments to the Author

Reviewer #3: Thank you for the opportunity given to me to peer review your work. Below is the feedback that I have after a review of your manuscript.

Overall, this manuscript provides a comprehensive overview of the HIV situation in Nigeria. The introduction effectively sets the stage by presenting global and national statistics, highlighting the key populations driving the epidemic, and emphasizing the importance of targeted interventions.

The methodology section is detailed and well-structured, explaining the data source, design, participants, sampling procedure, and statistical procedures.

The ethical considerations section adequately addresses the use of secondary data and emphasizes adherence to ethical guidelines during the original survey. However, considering that data was already approved, the ethical consideration section should be shorter.

For the figures depicting the percentage of households with at least one HIV-positive member (Figure 2) and HIV prevalence among participants by state (Figure 3) are informative and help to convey the regional disparities. However, it would be beneficial to include more detailed discussions or interpretations of these figures in the text.

Other suggestions for improvement

1. Discussion: While the paper provides a comprehensive set of results, the discussion section could be expanded to provide more in-depth interpretations of the findings. You can discuss the implications of the observed regional and gender disparities and how these insights can inform targeted interventions.

2. Clarity in Reporting: Some sentences are complex and could be simplified for clarity. Ensure that the language used is accessible to a broad audience, especially if the paper is intended for policymakers or healthcare practitioners.

3. Citations: Ensure that all statements and statistics are appropriately cited. While the paper generally provides sources for the data and information presented, a few statements could benefit from more specific citations.

Reviewer #4: The Manuscript covers secondary data covering 220000 individuals , 2 stage Clustering Sampling technique,, characterising gender ,Location ,Control, with Test ,Treat , Target at 95 % is praise worthy. Many references given on strategy for prevention , The author recommendation Has not been discussed from data analytics on biomedical, behavioural, and structural infrastructure and access to education for prevention of HIV .While awareness of HIV 58% mother were in urban ,29.8 % in rural ,and also when rural detects early(Page 18 L18,19), how rural prevalence more is surprising.

Geo Clustering to be defined by map for urban to claim the density with earlier reviewer recommendation.

NAIIS focus on 2018, But to maintain volatility of data , the trend in HIV updated conditions be mentioned.

The sources of spread of HIV and the rate of change in viral be highlighted for controlling the spread be discussed apart from prevalence . The high risk population has not been identified from ref 3 and 5 though separately mentioned as MTCT as primary measure. the effectiveness of Anti retro viral therapy be mentioned from referred literature. The procedure of Chi-square detection algorithm be explained for non-respondents . Table 2 and Table 3 are Unwieldy , and will be strenuous to researcher to assimilate. Therefore the scope to concise by presenting only the significant factors of p value below 0.05,and mentioning the event of the row that " does have HIV " only .The compliment event -" does not have HIV " - be removed - may be examined.

Page 3, Line 13 ,15 - !,1 million 2018 and 1.9million 2019 - rate of increase be mentioned

page 4 line 19 - MTCT is primary measure - literature of strategies be elucidated.

page 9 Line 5-15-49 years female -1.7% ,Male -0.8%- P vale significance commented

Page 9 - Line 11,12 -women 1.8% and men -1.0 % slightly higher be tested for statistical significance

Page 9 Line 21-Anti retro viral therapy-effectiveness of prevention from reference be discussed

Page 16 Line 5,6,7,8,- men , women , rural ,urban , before survey and after survey - the figures are to paraphrased in sentence properly

Page 16 line 23 , 24 - This sentence statistically significant be rephrased to distinguish between rural and urban .

page 18 Line 3,4,5 - not evidenced by data and recommended for removal

Page 18 Line 13,14, 15 -Limited access to education ,healthcare ,and resources be mentioned from literature and not evidenced from data .

Page 19 Line 4,5 - Reason for urban detection early is delayed due to patient load has not been evidenced with data .

Overall It is an informative progressive strength with the trend of spread rate that deserves publication.

7. PLOS authors have the option to publish the peer review history of their article (what does this mean?). If published, this will include your full peer review and any attached files.

**Do you want your identity to be public for this peer review?** For information about this choice, including consent withdrawal, please see our Privacy Policy.

Reviewer #3: No

Reviewer #4: **Yes: **A. Rajagopal

---

## [Decision Letter · Decision Letter 2]

6 Mar 2024

PGPH-D-23-02288R2

On Characterizing Gender and Locational Composition of Adult PLHIV in Nigeria: implications for HIV programming

Dear Lawal,

Thank you for submitting your manuscript to PLOS Global Public Health. After careful consideration, we feel that it has merit but does not fully meet PLOS Global Public Health’s publication criteria as it currently stands. Therefore, we invite you to submit a revised version of the manuscript that addresses the points raised during the review process.

We look forward to receiving your revised manuscript.

Kind regards,

Collins Otieno Asweto, PhD

Academic Editor

Journal Requirements:

2. Some material included in your submission may be copyrighted. According to PLOS’s copyright policy, authors who use figures or other material (e.g., graphics, clipart, maps) from another author or copyright holder must demonstrate or obtain permission to publish this material under the Creative Commons Attribution 4.0 International (CC BY 4.0) License used by PLOS journals. Please closely review the details of PLOS’s copyright requirements here: PLOS Licenses and Copyright. If you need to request permissions from a copyright holder, you may use PLOS's Copyright Content Permission form.

Potential Copyright Issues:

Figs 1&3: please (a) provide a direct link to the base layer of the map (i.e., the country or region border shape) and ensure this is also included in the figure legend; and (b) provide a link to the terms of use / license information for the base layer image or shapefile. We cannot publish proprietary or copyrighted maps (e.g. Google Maps, Mapquest) and the terms of use for your map base layer must be compatible with our CC-BY 4.0 license. 

"

Additional Editor Comments (if provided):

Reviewers' comments:

Reviewer's Responses to Questions

**Comments to the Author**

1. If the authors have adequately addressed your comments raised in a previous round of review and you feel that this manuscript is now acceptable for publication, you may indicate that here to bypass the “Comments to the Author” section, enter your conflict of interest statement in the “Confidential to Editor” section, and submit your "Accept" recommendation.

Reviewer #5: All comments have been addressed

Reviewer #6: (No Response)

2. Does this manuscript meet PLOS Global Public Health’s publication criteria? Is the manuscript technically sound, and do the data support the conclusions? The manuscript must describe methodologically and ethically rigorous research with conclusions that are appropriately drawn based on the data presented.

Reviewer #5: Yes

Reviewer #6: Yes

3. Has the statistical analysis been performed appropriately and rigorously?

Reviewer #5: Yes

Reviewer #6: Yes

4. Have the authors made all data underlying the findings in their manuscript fully available (please refer to the Data Availability Statement at the start of the manuscript PDF file)?

Reviewer #5: Yes

Reviewer #6: (No Response)

5. Is the manuscript presented in an intelligible fashion and written in standard English?

Reviewer #5: Yes

Reviewer #6: Yes

6. Review Comments to the Author

Reviewer #5: Firstly, I would like to commend the authors on the clear and structured presentation of your manuscript. The choice of Excel for data analysis is particularly noteworthy, as it makes the study accessible and replicable for a wide audience. Overall, the paper is well-written and organized efficiently, which enhances the reader's understanding and engagement with your work.

However, I have a few questions and suggestions that I believe could further strengthen the manuscript:

• Statistical Analysis (Page 8, Lines 2 and 4): The choice to employ both the Chi-square test and Fisher's exact test is appropriate for the analysis conducted. However, could the authors provide more detailed reasoning for the use of these specific tests in the context of your study? Elaborating on this choice would help readers understand the suitability and implications of these tests for the data and research objectives.

• Ethical Considerations (Line 12): The authors mention an ethics and regulatory board in their discussion on ethical considerations. Could they specify which board this refers to earlier in the text? The mention in lines 17 and 18 suggests a specific board, but clarity from the outset would help readers understand the ethical framework guiding the study. Additionally, considering this is a secondary analysis, could the authors clarify whether any form of Human subjects’ protection course or certification was completed by the research team? This information is crucial for readers to gauge the ethical rigor of your study.

• Results Section - Consideration of Rural vs. Urban Households: It is noted that 57% of the households in the study are in rural areas, leaving 43% in urban areas. This slight imbalance prompts a question regarding the potential impact on the statistical analysis. Have the authors considered applying weights to the data based on this rural-urban distribution? Although the current imbalance may not be significant, exploring this through weighted analysis could offer insights into whether it influences the results. This could be an interesting aspect to consider or discuss, even if only to validate the robustness of the findings.

In closing, I believe these additional details and clarifications could significantly enhance the manuscript's contribution to the field. The study presents valuable insights, and addressing these points would only strengthen its impact and reliability.

Kind regards,

Reviewer #6: 1. Manuscript number: PGPH-D-23-02288R2

2. Article title reviewed: On Characterizing Gender and Locational Composition of Adult PLHIV in Nigeria: implications for HIV programming.

3. Outputs of the review:

3.1. Review of the authors responses in specific comments

Overall, the authors have rigorously followed and integrated all of peer reviewers observations, except mainly the following point :

Comment 3 : Since the data used in the research was a secondary data, line 22,23, 24 should be talking about data extraction (ie. How your variables of interest were extracted from the Nigeria HIV/AIDS survey dataset).

Authors response : The authors did not extract data from the Nigeria HIV/AIDS survey dataset – only the technical report of the survey was assessed and data was manually extracted from it.

My conclusion about this point : This response is acceptable. But, the authors must include this additional information in the manuscript as reported here: the technical report of the survey was assessed and data was manually extracted from it.

3.2. My (review) comments

The manuscript provides a comprehensive overview of the HIV situation in Nigeria. However, in this manuscrit, hepatitis B and C are presented as opportunistic HIV infections (tables 2 and 3). We suggest these infections be considered as HIV co-infections. Regarding HIV opportunistic infections, we recommend to consider tuberculosis.

Unfortunately, this important study has not addressed HIV status among TB population due to the association exists between both infections. We know that TB is considered as the most common opportunistic infection among PLWH, and the prevalence HIV among TB population is often high particularly in African countries.

The same is true with the key populations (PS, MSM, IDUs, prisoners, etc.), whose prevalence is often higher than the national average in most African countries (Nigeria being no exception) and depending to the place of residence (rural/urban).

3.3. Recommandation :

It would be beneficial to address HIV status among TB population and among key populations (PS, MSM, IDUs, prisoners, etc.) based on the place of residence according to place of residence, as well as was done with the other groups in tables 2 and 3. Thus, if data are available, authors are invited to complete the tables 2 and 3 with the HIV situation in TB patients and in key populations. If data are not available, both situations should be discussed with evidence and references in the manuscript.

7. PLOS authors have the option to publish the peer review history of their article (what does this mean?). If published, this will include your full peer review and any attached files.

**Do you want your identity to be public for this peer review?** For information about this choice, including consent withdrawal, please see our Privacy Policy.

Reviewer #5: **Yes: **Erick Kiprotich Yegon

Reviewer #6: **Yes: **Lazare M'BOUNGOU

---

## [Decision Letter · Decision Letter 3]

4 Apr 2024

PGPH-D-23-02288R3

On Characterizing Gender and Locational Composition of Adult PLHIV in Nigeria: implications for HIV programming

Dear Lawal,

Thank you for submitting your manuscript to PLOS Global Public Health. After careful consideration, we feel that it has merit but does not fully meet PLOS Global Public Health’s publication criteria as it currently stands. Therefore, we invite you to submit a revised version of the manuscript that addresses the points raised during the review process.

We look forward to receiving your revised manuscript.

Kind regards,

Collins Otieno Asweto, PhD

Academic Editor

Journal Requirements:

Additional Editor Comments (if provided):

Reviewers' comments:

Reviewer's Responses to Questions

**Comments to the Author**

1. If the authors have adequately addressed your comments raised in a previous round of review and you feel that this manuscript is now acceptable for publication, you may indicate that here to bypass the “Comments to the Author” section, enter your conflict of interest statement in the “Confidential to Editor” section, and submit your "Accept" recommendation.

Reviewer #3: (No Response)

Reviewer #5: All comments have been addressed

Reviewer #7: (No Response)

Reviewer #8: All comments have been addressed

2. Does this manuscript meet PLOS Global Public Health’s publication criteria? Is the manuscript technically sound, and do the data support the conclusions? The manuscript must describe methodologically and ethically rigorous research with conclusions that are appropriately drawn based on the data presented.

Reviewer #3: Yes

Reviewer #5: Yes

Reviewer #7: Partly

Reviewer #8: Yes

3. Has the statistical analysis been performed appropriately and rigorously?

Reviewer #3: Yes

Reviewer #5: Yes

Reviewer #7: No

Reviewer #8: Yes

4. Have the authors made all data underlying the findings in their manuscript fully available (please refer to the Data Availability Statement at the start of the manuscript PDF file)?

Reviewer #3: Yes

Reviewer #5: Yes

Reviewer #7: Yes

Reviewer #8: Yes

5. Is the manuscript presented in an intelligible fashion and written in standard English?

Reviewer #3: Yes

Reviewer #5: Yes

Reviewer #7: Yes

Reviewer #8: Yes

6. Review Comments to the Author

Reviewer #3: The ethical considerations, as well as the detailed interpretation of figures 2 and 3, were not addressed in this version. However, there was some discussion regarding the implications of observed regional and gender disparities, albeit only partially addressing how these insights can inform targeted interventions. This suggests a need for addressing ethical consideration comments in the first review as well as a deeper dive into the significance of the data presented in figures 2 and 3 to enhance the understanding of potential interventions aimed at addressing disparities.

Reviewer #5: All my comments have been addressed sufficiently and the manuscript is now in a better shape and format for acceptance.

Reviewer #7: Summary

The authors have demonstrated a good understanding of the subject matter. Conclusions ha ve also been reached covering key areas of the study objective. However, based on the interest of the study which essentially examined the distribution of HIV viz-a-viz its prevalence, diagnosis etc. across gender and location, analyses were basically descriptive and these have been well established. This limits the contribution of the study to literature. However, extending the study focus to examining causative factors particularly in respect of contexts/locations can improve the quality of the study.

Recommendation

Publish with major revision

Review feedback

Page 2

Lines 7/11: Authors omitted sample size in the method section

Line 9: “These groups”: Authors may want to first, clearly identify the different groups

Lines 13/22: Evidence of statistical significance of difference between rural and urban not shown

Overall, it appeared the only location of focus, as shown in the abstract, was the rural/urban dichotomy. Such finding appear evident in literature already

Page 3

Lines 11/13: Authors may want to break sentences to smaller parts to improve clarity on the origin of HIV, its prevalence in sub-Saharan Africa and the current situation in the sub-region

Page 5

Lines 3/8: The paragraph appeared to have addressed the core focus of the study.

Lines 9/17: The strength of the study appeared to be the combined effect of gender and location as well as the use of a nationally representative study. However the analytic method adopted appeared to be insufficient to address the objective. In particular, gender disparities among PLHIV have been well established, perhaps a focus on locational clustering and extension of analysis beyond bivariate level can further strengthen the justification for the study.

Page 6

Line 5: While the sampling procedure for the survey was described extensively, specific study sample size was omitted by the Authors.

Line 11: Perhaps a justification for selected age range is essential

Page 7

Lines 20/23: Nice attempt but I wonder if more accuracy of sample can be achieved using the actual raw survey data, rather than the estimates derived from reported percentages and total sample size

Page 8

Line 6: A 10% significant level was used, unless of course there are justifiable arguments, the robustness of the analysis to detect true significance is reduced. The chances of committing a type 1 error is increased such that a true null hypothesis is rejected. Wrong decision might have been taken on the significance of differences in examining gender and location differences. I recommend testing the clustering at, at least, two different lower significant levels.

Page 9

Line 1: Authors wrote 58% rural residents in the table but reported 57%

Line 7: Authors did not show sample size on Table 1. The practice is to either show the denominator (N) at the top of the table if the total sample for each variable in the table is the same or show total sample for each of the variable in the table

Page 10

Significant levels of Table 2 were actually 0.1%, rather than the earlier 10% stated by the Authors

Reviewer #8: The authors took into consideration the comments made by the reviewers, and made the appropriate corrections.

7. PLOS authors have the option to publish the peer review history of their article (what does this mean?). If published, this will include your full peer review and any attached files.

**Do you want your identity to be public for this peer review?** For information about this choice, including consent withdrawal, please see our Privacy Policy.

Reviewer #3: No

Reviewer #5: **Yes: **Erick Kiprotich Yegon

Reviewer #7: No

Reviewer #8: No

---

## [Decision Letter · Decision Letter 4]

5 Jun 2024

PGPH-D-23-02288R4

On Characterizing Gender and Locational Composition of Adult PLHIV in Nigeria: implications for HIV programming

Dear Dr. Lawal,

Thank you for submitting your manuscript to PLOS Global Public Health. After careful consideration, we feel that it has merit but does not fully meet PLOS Global Public Health’s publication criteria as it currently stands. Therefore, we invite you to submit a revised version of the manuscript that addresses the points raised during the review process.

The manuscript has been re-evaluated by two of the previous reviewers. Although reviewer 3 has not requested any further changes, reviewer 7 notes several issues - please see the comments below.

Could you please carefully revise the manuscript to address all comments raised?

We look forward to receiving your revised manuscript.

Kind regards,

Steve Zimmerman, PhD

PLOS Staff Editor

Journal Requirements:

2. Some material included in your submission may be copyrighted. According to PLOS’s copyright policy, authors who use figures or other material (e.g., graphics, clipart, maps) from another author or copyright holder must demonstrate or obtain permission to publish this material under the Creative Commons Attribution 4.0 International (CC BY 4.0) License used by PLOS journals. Please closely review the details of PLOS’s copyright requirements here: PLOS Licenses and Copyright. If you need to request permissions from a copyright holder, you may use PLOS's Copyright Content Permission form.

Potential Copyright Issues:

Figs 1&3: please (a) provide a direct link to the base layer of the map (i.e., the country or region border shape) and ensure this is also included in the figure legend; and (b) provide a link to the terms of use / license information for the base layer image or shapefile. We cannot publish proprietary or copyrighted maps (e.g. Google Maps, Mapquest) and the terms of use for your map base layer must be compatible with our CC-BY 4.0 license. 

Additional Editor Comments (if provided):

Reviewers' comments:

Reviewer's Responses to Questions

**Comments to the Author**

1. If the authors have adequately addressed your comments raised in a previous round of review and you feel that this manuscript is now acceptable for publication, you may indicate that here to bypass the “Comments to the Author” section, enter your conflict of interest statement in the “Confidential to Editor” section, and submit your "Accept" recommendation.

Reviewer #3: (No Response)

Reviewer #7: All comments have been addressed

2. Does this manuscript meet PLOS Global Public Health’s publication criteria? Is the manuscript technically sound, and do the data support the conclusions? The manuscript must describe methodologically and ethically rigorous research with conclusions that are appropriately drawn based on the data presented.

Reviewer #3: (No Response)

Reviewer #7: Yes

3. Has the statistical analysis been performed appropriately and rigorously?

Reviewer #3: (No Response)

Reviewer #7: Yes

4. Have the authors made all data underlying the findings in their manuscript fully available (please refer to the Data Availability Statement at the start of the manuscript PDF file)?

Reviewer #3: (No Response)

Reviewer #7: Yes

5. Is the manuscript presented in an intelligible fashion and written in standard English?

Reviewer #3: (No Response)

Reviewer #7: Yes

6. Review Comments to the Author

Reviewer #3: (No Response)

Reviewer #7: ABSTRACT

3 This study CHARACTERIZED the gender and locational variations in the spread and control of HIV among adult Persons Living with HIV (PLHIV) to identify the clustering of PLHIV in Nigeria

INTRODUCTION

Page 4

9 2025, which is essential for the detection and prevention of mother-to-child transmission of

HIV (PMTCT) before pregnancy among reproductive age.

Page 7

2 “Prevalence of HIV” I think is reported in proportion, not frequency

6 Self-reported HIV testing: Number of participants who REPORTED ever receivING an HIV test result and receivING an HIV test result. I presume the REPORTED part was omitted.

RESULTS

Page 9

3 Rural population proportion is 58% as shown in Table 1 (This was not updated)

Page 10

4 Chi square and Fisher exact tests are either testing for association or independence. Based on the objective of this manuscript, the tests are for association, not difference as reported across the manuscript. Reporting difference could suggest that tests of mean difference have been conducted. That can be misleading. I would advise that Authors report the distributional proportions. For example, rather than saying “there is a significant difference”, I would say ”there is a significant variation in the distribution…”

5 Reporting p-values as being less than 0.001 suggests you are comparing your estimates’ p-values with 0.1% level of significance (These have not been updated)

DISCUSSION

Page 18

6 I would advise hat Authors begin the discussion section with the strength of this study. First few paragraphs should focus on discussing findings from this study (rather than those from NAIIS) and relate to findings from literature including NAIIS report.

General grammatical checks would also improve the manuscript

7. PLOS authors have the option to publish the peer review history of their article (what does this mean?). If published, this will include your full peer review and any attached files.

**Do you want your identity to be public for this peer review?** For information about this choice, including consent withdrawal, please see our Privacy Policy.

Reviewer #3: No

Reviewer #7: No

---

## [Decision Letter · Decision Letter 5]

26 Jul 2024

PGPH-D-23-02288R5

On Characterizing Gender and Locational Composition of Adult PLHIV in Nigeria: implications for HIV programming

Dear Dr. Lawal,

Thank you for submitting your manuscript to PLOS Global Public Health. After careful consideration, we feel that it has merit but does not fully meet PLOS Global Public Health’s publication criteria as it currently stands. Therefore, we invite you to submit a revised version of the manuscript that addresses the points raised during the review process.

I appreciate that this has been a long process, and thank the authors and reviewers for their patience. There are just a few minor concerns to address before the manuscript can be ready for publication, which have been raised by reviewer 7, below. However, I do not agree with the concern regarding the p values. Our guidelines state "Report exact p-values for all values greater than or equal to 0.001. P-values less than 0.001 may be expressed as p < 0.001", and that is what the authors have done. I can also see that you have clarified the value of alpha in your Methods. You should, however, also clarify the meaning of "*" in your tables.

We look forward to receiving your revised manuscript.

Kind regards,

Marianne Clemence

Staff Editor

Journal Requirements:

Additional Editor Comments (if provided):

Reviewers' comments:

Reviewer's Responses to Questions

**Comments to the Author**

1. If the authors have adequately addressed your comments raised in a previous round of review and you feel that this manuscript is now acceptable for publication, you may indicate that here to bypass the “Comments to the Author” section, enter your conflict of interest statement in the “Confidential to Editor” section, and submit your "Accept" recommendation.

Reviewer #3: All comments have been addressed

Reviewer #7: (No Response)

2. Does this manuscript meet PLOS Global Public Health’s publication criteria? Is the manuscript technically sound, and do the data support the conclusions? The manuscript must describe methodologically and ethically rigorous research with conclusions that are appropriately drawn based on the data presented.

Reviewer #3: Yes

Reviewer #7: Yes

3. Has the statistical analysis been performed appropriately and rigorously?

Reviewer #3: Yes

Reviewer #7: Yes

4. Have the authors made all data underlying the findings in their manuscript fully available (please refer to the Data Availability Statement at the start of the manuscript PDF file)?

Reviewer #3: Yes

Reviewer #7: Yes

5. Is the manuscript presented in an intelligible fashion and written in standard English?

Reviewer #3: Yes

Reviewer #7: Yes

6. Review Comments to the Author

Reviewer #3: The author has addressed my comments and the work is presented in standard english.

Reviewer #7: Generally, while authors have improved the manuscript draft based on some suggestions earlier highlighted, other suggestions are yet to be accommodated. An instance is the significance level which still appears as though analysis was done using 10% significant level. I would implore authors to adopt “association” rather than “difference” used in the reporting. This can be misleading as Chi-square/Fishers’ tests used here are tests of association, not of difference. For example, report can be written as: “There is a significant association between location and HIV prevalence as higher prevalence was observed among rural residents compared to those in urban areas”.

Page 4

Lines 5-7: At the end of this sentence, “The aim is to guarantee that 90% …”, the “respectively” appears to suggest something is missing. While the two milestone dates were stated, the events to which they individually refer are not clear.

Page 9:

Update Table 1 where rural percent is 57% as written on Page 9, Line 3. I think the rural-urban divide percentages needs to be updated

In Tables 2 and 3, it appears that the authors still tested the associations between the outcomes and Gender and locational variable at 10% (0.001) rather than the 5%(0.05) earlier mentioned. This should be updated in both tables and corresponding reports. If these tests had been done at 5% level, it would be more appropriate to compare tests p-value to the significant level i.i p<0.05 rather than p<0.001.

Page 19:

Line 4: “reported by the as the …”, something appears to be missing in this sentence

Page 22:

Lines 8-10: The strength of this study can be better represented if specific limitations can be mentioned rather than the generic sample size, sampling methodology and measurement validity mentioned. NAIIS is a nationally representative data which incorporated rigorous method design, the extent to which earlier mentioned limitations affect representativeness of this study and its generalizability should be stated and clarified.

7. PLOS authors have the option to publish the peer review history of their article (what does this mean?). If published, this will include your full peer review and any attached files.

**Do you want your identity to be public for this peer review?** For information about this choice, including consent withdrawal, please see our Privacy Policy.

Reviewer #3: No

Reviewer #7: No

---

## [Editor Report · Decision Letter 6]

7 Aug 2024

On Characterizing Gender and Locational Composition of Adult PLHIV in Nigeria: implications for HIV programming

PGPH-D-23-02288R6

Dear Mr. Lawal,

We are pleased to inform you that your manuscript 'On Characterizing Gender and Locational Composition of Adult PLHIV in Nigeria: implications for HIV programming' has been provisionally accepted for publication in PLOS Global Public Health.

Best regards,

Julia Robinson

Executive Editor